# SIGNAL CODING AND RECONSTRUCTION USING SPIKE TRAINS

## ABSTRACT

In many animal sensory pathways, the transformation from external stimuli to spike trains is essentially deterministic. In this context, a new mathematical framework for coding and reconstruction, based on a biologically plausible model of the spiking neuron, is presented. The framework considers encoding of a signal through spike trains generated by an ensemble of neurons via a standard convolve-then-threshold mechanism, albeit with a wide variety of convolution kernels. Neurons are distinguished by their convolution kernels and threshold values. Reconstruction is posited as a convex optimization minimizing energy. Formal conditions under which perfect and approximate reconstruction of the signal from the spike trains is possible are then identified. Coding experiments on a large audio dataset are presented to demonstrate the strength of the framework.

## 1 INTRODUCTION

In biological systems, sensory stimuli is communicated to the brain primarily via ensembles of discrete events that are spatiotemporally compact electrical disturbances generated by neurons, otherwise known as spikes. Spike train representation of signals, when sparse, are not only intrinsically energy efficient, but can also facilitate downstream computation(6; 10). In their seminal work, Olshausen and Field (13) showed how efficient codes can arise from learning sparse representations of natural stimulus statistics, resulting in striking similarities with observed biological receptive fields. (19) developed a biophysically motivated spiking neural network which for the first time predicted the full diversity of V1 simple cell receptive field shapes when trained on natural images. Although these results signify substantial progress, an effective end to end signal processing framework that deterministically represents signals via spike train ensembles is yet to be laid out. Here we present a new framework for coding and reconstruction leveraging a biologically plausible coding mechanism which is a superset of the standard leaky integrate-and-fire neuron model (5).

Our proposed framework identifies reconstruction guarantees for a very general class of signals—those with *finite rate of innovation* (18)—as shown in our perfect and approximate reconstruction theorems. Most other classes, e.g. bandlimited signals, are subsets of this class. The proposed technique first formulates reconstruction as an optimization that minimizes the energy of the reconstructed signal subject to consistency with the spike train, and then solves it in closed form. We then identify a general class of signals for which reconstruction is provably perfect under certain ideal conditions. Subsequently, we present a mathematical bound on the error of an approximate reconstruction when the model deviates from those ideal conditions. Finally, we present simulation experiments coding for a large dataset of audio signals that demonstrate the efficacy of the framework. In a separate set of experiments on a smaller subset of audio signals we compare our framework with existing sparse coding algorithms viz matching pursuit and orthogonal matching pursuit, establishing the strength of our technique.

The remainder of the paper is structured as follows. In Sections 2 and 3 we introduce the coding and decoding frameworks. Section 4 identifies the class of signals for which perfect reconstruction is achievable if certain ideal conditions are met. In Section 5 we discuss how in practice those ideal conditions can be approached and provide a mathematical bound for approximate reconstruction. Simulation results are presented in Section 6. We conclude in Section 8.

## 2 CODING

The general class of deterministic mappings (i.e., the set of all nonlinear operators) from continuous time signals to spike trains is difficult to characterize because the space of all spike trains does not lend itself to a natural topology that is universally embraced. The result is that simple characterizations, such as the set of all continuous operators, can not be posited in a manner that has general consensus. To resolve this issue, we take a cue from biological systems. In most animal sensory pathways, external stimulus passes through a series of transformations before being turned into spike trains(17). For example, visual signal in the retina is processed by multiple layers of non-spiking horizontal, amacrine and bipolar cells, before being converted into spike trains by the retinal ganglion cells. Accordingly, we can consider the set of transformations that pass via an intermediate continuous time signal which is then transformed into a spike train through a stereotyped mapping where spikes mark threshold crossings. The complexity of the operator now lies in the mapping from the continuous time input signal to the continuous time intermediate signal. Since any time invariant, continuous, nonlinear operator with fading memory can be approximated by a finite Volterra series operator(2), this general class of nonlinear operators from continuous time signals to spike trains can be modeled as the composition of a finite Volterra series operator and a neuronal thresholding operation to generate a spike train. Here, the simplest subclass of these transformations is considered: the case where the Volterra series operator has a single causal, bounded-time, *linear* term, the output of which is composed with a thresholding operation of a potentially time varying threshold. The overall operator from the input signal to the spike train remains *nonlinear* due to the thresholding operation. The code generated by an ensemble of such transformations, corresponding to an ensemble of spike trains, is explored.

Formally, we assume the input signal $X(t)$ to be a *bounded square integrable function* over the compact interval $[0, T]$ for some $T \in \mathbb{R}^+$, i.e., we are interested in the class of input signals $\mathcal{F} = \{X(t)|X(t) \in L^2[0,T]\}$. Since the framework involves signal snippets of arbitrary length, this choice of $T$ is without loss of generalization. We assume an ensemble of convolution kernels $K = \{K^j | j \in Z^+, j \leq n\}$, consisting of $n$ kernels $K^j, j = 1, \ldots, n$. We assume that $K^j(t)$ is a continuous function on a bounded time interval $[0, T]$, i.e. $\forall j \in \{1, \ldots, n\}, K^j(t) \in C[0, T]$, $T \in \mathbb{R}^+$. Finally, we assume that $K^j$ has a time varying threshold denoted by $T^j(t)$.

The ensemble of convolution kernels $K$ encodes a given input signal $X(t)$ into a sequence of spikes $\{(t_i, K^{j_i})\}$, where the $i^{th}$ spike is produced by the $j_i^{th}$ kernel $K^{j_i}$ at time $t_i$ if and only if: $\int X(\tau)K^{j_i}(t_i - \tau)d\tau = T^{j_i}(t_i)$ In our experiments a specific threshold function is assumed in which the time varying threshold $T^j(t)$ of the $jth$ kernel remains constant at $C^j$ until that kernel produces a spike, at which time an *after-hyperpolarization potential (ahp)* is introduced to raise the threshold to a high value $M^j \gg C^j$, which then drops back linearly to its original value within a refractory period $\delta_j$. Stated formally,

$$T^j(t) = \begin{cases} C^j, & t - \delta_j > t_l^j(t) \\ M^j - \frac{(t - t_l^j(t))(M^j - C^j)}{\delta_j}, & t - \delta_j \leq t_l^j(t) \end{cases} \tag{1}$$

Where $t_l^j(t)$ denotes the time of the last spike generated by $K^j$ prior to time $t$.

## 3 DECODING

How rich is the coding mechanism just described? We can investigate this question formally by positing a decoding module. The objective of the decoding module is to reconstruct the original signal from the encoded ensemble of spike trains. It is worthwhile to mention that to be able to communicate signals properly by our proposed framework, the decoding module needs to be designed in a manner so that it can operate solely on the spike train data handed over by the encoding module, without explicit access to the input signal itself. Considering the prospect of the invertibility of the coding scheme, we seek a signal that satisfies the same set of constraints as the original signal when generating all spikes apropos the set of kernels in ensemble $K$. Recognizing that such a signal might not be unique, we choose the reconstructed signal as the one with minimum $L2$-norm.

Formally, the reconstruction (denoted by $X^*(t)$) of the input signal $X(t)$ is formulated to be the solution to the optimization problem:

$$X^*(t) = \underset{\tilde{X}}{\mathrm{argmin}} ||\tilde{X}(t)||_2^2$$

$$\text{s.t. } \int \tilde{X}(\tau) K^{j_i}(t_i - \tau) d\tau = T^{j_i}(t_i); 1 \le i \le N \tag{2}$$

where $\{(t_i, K^{j_i}) | i \in \{1, ..., N\}\}$ is the set of all spikes generated by the encoder. The choice of $L^2$ minimization as the objective of the reconstruction problem—which is the linchpin of our framework, as demonstrated in the theorems—can only be weakly justified at the current juncture. The perfect reconstruction theorem that follows provides the strong justification. As it stands, the $L^2$ minimization objective is in congruence with the dictum of energy efficiency in biological systems. The assumption is that, of all signals, the one with the minimum energy that is consistent with the spike trains is desirable. Additionally, an $L^2$ minimization in the objective of (2) reduces the convex optimization problem to a solvable linear system of equations as shown in Lemmas 1 and 3. Later we shall show that $L^2$-minimization has the surprising benefit of recovering the original signal perfectly under certain conditions.

## 4 SIGNAL CLASS FOR PERFECT RECONSTRUCTION

To establish the effectiveness of the described coding-decoding model, we have to evaluate the accuracy of reconstruction over a class of input signals. We observe that in general the encoding of square integrable signals into spike trains is not a one-to-one map; the same set of spikes can be generated by different signals so as to result in the same convolved values at the spike times. Naturally, with a finite and fixed ensemble of kernels $K$, one cannot achieve perfect reconstruction for the general class of signals $\mathcal{F}$ as defined in Section 2. We now restrict ourselves to a subset $\mathcal{G}$ of the original class $\mathcal{F}$ defined as $\mathcal{G} = \{X(t)|X(t) \in \mathcal{F}, X(t) = \sum_{p=1}^{N} \alpha_p K^{j_p}(t_p - t), j_p \in \{1, ..., n\}, \alpha_p \in \mathbb{R}, t_p \in \mathbb{R}^+, N \in Z^+\}$ and address the question of reconstruction accuracy. Essentially $\mathcal{G}$ consists of all linear combinations of arbitrarily shifted kernel functions. $N$ is bounded above by the total number of spikes that the ensemble $K$ can generate over $[0, T]$. In the parlance of signal processing, $\mathcal{G}$ constitutes *Finite rate of Innovation* signals (18). For the class $\mathcal{G}$ the *perfect reconstruction theorem* is presented below. The theorem is proved with the help of three lemmas.

**Perfect Reconstruction Theorem:** Let $X(t) \in \mathcal{G}$ be an input signal. Then for appropriately chosen time-varying thresholds of the kernels, the reconstruction, $X^*(t)$, resulting from the proposed coding-decoding framework is accurate with respect to the $L^2$ metric, i.e., $||X^*(t) - X(t)||_2 = 0$.

**Lemma 1:** The solution $X^*(t)$ to the reconstruction problem given by (2) can be written as:

$$X^*(t) = \sum_{i=1}^{N} \alpha_i K^{j_i}(t_i - t) \tag{3}$$

where the coefficients $\alpha_i \in \mathbb{R}$ can be solved from a system of linear equations.

**Proof:** An approach analogous to the Representer Theorem (15), splitting a putative solution to (2) into its within the span of the kernels component and a remnant orthogonal component, results in equation (3). In essence, the reconstructed signal $X^*(t)$ becomes a summation of the kernels, shifted to their respective times of generation of spikes, scaled by appropriate coefficients. Plugging (3) into the constraints (2) gives:

$$\forall_{1 \le i \le N}; \int \sum_{k=1}^{N} \alpha_k K^{j_k}(t_k - t) K^{j_i}(t_i - t) d\tau = T^{j_i}(t_i)$$

Setting $b_i = T^{j_i}(t_i)$ and $P_{ik} = \int K^{j_k}(t_k - \tau) K^{j_i}(t_i - \tau) d\tau$ results in:

$$\forall_{1 \le i \le N}; \sum_{k=1}^{N} P_{ik} \alpha_k = b_i \tag{4}$$

Equation (4) defines a system of $N$ equations in $N$ unknowns of the form:

$$P\alpha = T \tag{5}$$

where $\alpha = \langle \alpha_1, ..., \alpha_N \rangle^T$, $T = \langle T^{j_1}(t_1), ..., T^{j_N}(t_N) \rangle^T$ and $P$ is an $N \times N$ matrix with elements $P_{ik} = \int K^{j_k}(t_k - \tau) K^{j_i}(t_i - \tau) d\tau$. Clearly $P$ is the Gramian Matrix of the shifted kernels

$\{K^{j_i}(t_i - t) | i \in 1, 2, ..., N\}$ in the Hilbert space with the standard inner product. It is well known that $P$ is invertible if and only if $\{K^{j_i}(t_i - t) | i \in 1, 2, ..., N\}$ is a linearly independent set. If $P$ is invertible $\alpha$ has a unique solution. If, on the other hand, $P$ is not invertible, $\alpha$ has multiple solutions. However, as the next lemma shows, every such solution leads to the same reconstruction $X^*(t)$, and hence any value of $\alpha$ that satisfies 5 can be chosen. We note in passing that in our experiments we have used the least square solution. $\qquad\square$

**Import:** The goal of the optimization problem is to find the best object in the feasible set. However, the application of the Representer Theorem converts the constraints into a determined system of unknowns and equations, turning the focus onto the feasible set, effectively changing the optimization problem into a solvable system that results in a closed form solution for the $\alpha_i$'s. This implies that instead of solving (2), we can solve for the reconstruction from $X^*(t) = \sum_{i=1}^{N} \alpha_i K^{j_i}(t_i - t)$, where $\alpha_i$ is the i-th element of $\alpha = P^{-1}T$. Here, $P^{-1}$ represents either the inverse or the Moore-Penrose inverse, as the case may be.

**Lemma 2:** Let equation 5 resulting from the optimization problem 2 have multiple solutions. Consider any two different solutions for $\alpha$, namely $\alpha_1$ and $\alpha_2$, and hence the corresponding reconstructions are given by $X_1(t) = \sum_{i=1}^{N} \alpha_{1i} K^{j_i}(t_i - t)$ and $X_2(t) = \sum_{i=1}^{N} \alpha_{2i} K^{j_i}(t_i - t)$, respectively. Then $X_1 = X_2$.

**Proof:** The proof of this lemma follows from the existence of a unique function in the Hilbert Space spanned by $\{K^{j_i}(t_i - t) | i \in 1, 2, ..., N\}$ that satisfies the constraint of equation 2. The details of the proof is furnished in the appendix A.

**Import:** Lemma 2 essentially establishes the uniqueness of solution to the optimization problem formulated in 2 as any solution to equation 5. The proof follows from the fact that the reconstruction is in the span of the shifted kernels $\{K^{j_i}(t_i - t) | i \in 1, 2, ..., N\}$ and the inner products of the reconstruction with each of $K^{j_i}(t_i - t)$ is given (by the spike constraints of 2). Such a reconstruction must be unique in the subspace $S$.

**Lemma 3:** Let $X^*(t)$ be the reconstruction of an input signal $X(t)$ and $\{(t_i, K^{j_i})\}_{i=1}^{N}$ be the set of spikes generated. Then, for any arbitrary signal $\tilde{X}(t)$ within the span of $\{K^{j_i}(t_i - t) | i \in \{1, 2, ..., N\}\}$, i.e., the set of shifted kernels at respective spike times, given by $\tilde{X}(t) = \sum_{i=1}^{N} a_i K^{j_i}(t_i - t)$ the following inequality holds: $||X(t) - X^*(t)|| \leq ||X(t) - \tilde{X}(t)||$

**Proof:**

$$||X(t) - \tilde{X}(t)|| = || \underbrace{X(t) - X^*(t)}_{A} + \underbrace{X^*(t) - \tilde{X}(t)}_{B} ||$$

First, $\langle A, K^{j_i}(t_i - t) \rangle = \langle X(t), K^{j_i}(t_i - t) \rangle - \langle X^*(t), K^{j_i}(t_i - t) \rangle, \forall i \in \{1, 2, .., N\}$

$$= T^{j_i}(t_i) - T^{j_i}(t_i) = 0 \qquad \text{(Using the constraints in (2) \& (2))}$$

Second, $\langle A, B \rangle = \langle A, \sum_{i=1}^{N} (\alpha_i - a_i) K^{j_i}(t_i - t) \rangle \qquad$ (By Lemma 1 $X^*(t) = \sum_{i=1}^{N} \alpha_i K^{j_i}(t_i - t)$)

$$= \sum_{i=1}^{N} (\alpha_i - a_i) \langle A, K^{j_i}(t_i - t) \rangle = 0$$

$$\implies ||X(t) - \tilde{X}(t)||^2 = ||A + B||^2 = ||A||^2 + ||B||^2 \geq ||A||^2 = ||X(t) - X^*(t)||^2$$

$$\implies ||X(t) - \tilde{X}(t)|| \geq ||X(t) - X^*(t)|| \qquad\qquad\qquad\qquad\qquad\square$$

**Import:** The implication of the above lemma is quite remarkable. The objective defined in (2) chooses a signal with minimum energy satisfying the constraints, deemed the reconstructed signal. However as the lemma demonstrates, this signal also has the minimum *error* with respect to the input signal in the span of the shifted kernels. This signifies that our choice of the objective in the decoding module not only draws from biologically motivated energy optimization principles, but also performs optimally in terms of reconstructing the original input signal within the span of the appropriately shifted spike generating kernels.

**Corollary:** An important consequence of Lemma 3 is that *additional spikes in the system do not worsen the reconstruction*. For a given input signal $X(t)$ if $S_1$ and $S_2$ are two sets of spike trains where $S_1 \subset S_2$, the second a superset of the first, then Lemma 3 implies that the reconstruction due to $S_2$ is at least as good as the reconstruction due to $S_1$ because the reconstruction due to $S_1$ is in the span of the shifted kernel functions of $S_2$ as $S_1 \subset S_2$. This immediately leads to the conclusion that

for a given input signal the more kernels we add to the ensemble the better the reconstruction.

**Proof of the Theorem:** The proof of the theorem follows directly from Lemma 3. Since the input signal $X(t) \in \mathcal{G}$, let $X(t)$ be given by: $X(t) = \sum_{p=1}^{N} \alpha_p K^{j_p}(t_p - t)$ $(\alpha_p \in \mathbb{R}, t_p \in \mathbb{R}^+, N \in Z^+)$ Assume that the time varying thresholds of the kernels in our kernel ensemble $K$ are set in such a manner that the following conditions are satisfied: $\langle X(t), K^{j_p}(t_p - t) \rangle = T^{j_p}(t_p)$ $\forall p \in \{1, ..., N\}$ i.e., each of the kernels $K^{j_p}$ at the very least produces a spike at time $t_p$ against $X(t)$ (regardless of other spikes at other times). Clearly then $X(t)$ lies in the span of the appropriately shifted response functions of the spike generating kernels. Applying Lemma 3 it follows that: $||X(t) - X^*(t)||_2 \leq ||X(t) - X(t)||_2 = 0$ □

**Import:** In addition to demonstrating the potency of the coding-decoding scheme, this theorem frames Barlow's efficient coding hypothesis (1)—that the coding strategy of sensory neurons be adapted to the statistics of the stimuli—in mathematically concrete terms. Going by the theorem, the spike based encoding necessitates the signals to be in the span of the encoding kernels for perfect reconstruction. Inverting the argument, kernels must learn to adapt to the *basis* elements that generate the signal corpora for superior reconstruction.

# 5 APPROXIMATE RECONSTRUCTION AND THE EFFECT OF AHP

The perfect reconstruction theorem stipulates the conditions under which exact recovery of a signal is feasible in the proposed framework. At first glance, it may seem challenging to meet these conditions for an arbitrary class of natural signals. The concern stems from two difficulties: firstly, given a fixed set of kernels, the input signal may not lie in the span of their arbitrary shifts, and secondly, we may not be able to generate spikes at the desired locations as postulated in the proof of the theorem.

To address these issues, we observe that our decoding model is a continuous transformation from the space of spike trains to $L^2$-functions, in the sense that small changes in spike times or a slight mismatch of the spiking kernels from the components of the signal, bring about only small changes in the reconstruction. In what follows, we furnish an *Approximate Reconstruction Theorem* (C) that provides a bound on the reconstruction error under such deviations. To address the first problem, it is important to choose kernel functions appropriately so that they can represent the input signals reasonably well. One can leverage biological knowledge; for example, it is well-known that auditory filters are effectively modeled using gammatones (14). Hence our experiments in Section 6 on auditory signals were coded using gammatone kernels. Not surprisingly, the reconstructions were excellent. To alleviate the second problem, we observe that spikes can be produced reasonably close to the desired locations by setting a low baseline threshold and a small refractory period of the *after-hyperpolarization potential (ahp)* for each kernel, a technique that is guaranteed to give good results as is confirmed by our experiments in Section 6. The following lemma formalizes the notion of how lowering the threshold and the refractory period of a kernel helps in generating spikes at the desired locations. The lemma is followed by the *Approximate Reconstruction Theorem*.

**Lemma 4:** Let $X(t)$ be an input signal. Let $K^p$ be a kernel for which we want to generate a spike at time $t_p$. Let the inner product $\langle X(t), K^p(t_p - t) \rangle = I^p$. Then, if the baseline threshold of the kernel $K^p$ is $C^p \leq I^p$ and the absolute refractory period is $\delta$ as modeled in Equation 1, the kernel $K^p$ must produce a spike in the interval $[t_p - \delta, t_p]$ according to the threshold model defined in Equation 1.

**Proof:** The proof of this lemma follows directly from the *intermediate value theorem* and is detailed in appendix B.

**Approximate Reconstruction Theorem:** Let the following assumptions be true:

• $X(t)$, the input signal to the proposed framework, can be written as a linear combination of some component functions as- $X(t) = \sum_{i=1}^{N} \alpha_i f_{p_i}(t_i - t)$ where $\alpha_i$ are bounded real coefficients, the component functions $f_{p_i}(t)$ are chosen from a possibly infinite set $\mathcal{H} = \{f_i(t) | i \in Z^+, ||f_i(t)|| = 1\}$ of functions of unit $L^2$ norm with compact support, and the corresponding $t_i \in R^+$ are chosen to be bounded arbitrary time shifts of the component functions so that the overall signal has a compact support in $[0, T]$ for some $T \in R^+$ and thus the input signal still belongs to the same class of signals $\mathcal{F}$, as defined in section 2.

• There is at least one kernel $K^{j_i}$ from the bag of encoding kernels K, such that the $L^2$-distance of $f_{p_i}(t)$ from $K^{j_i}(t)$ is bounded. Formally, $\exists \delta \in R^+$ s.t. $||f_{p_i}(t) - K^{j_i}(t)||_2 < \delta \ \forall i \in \{1, ..., N\}$.

• When $X(t)$ is encoded by the proposed framework, each one of these kernels $K^{j_i}$ produce a spike at time $t_i'$ at threshold $T_i$ such that $|t_i - t_i'| < \Delta \ \forall i$, for some $\Delta \in R^+$.

• Each kernel $K^j \in K$ satisfies a Lipschitz type condition as follows:

$\exists\, C \in R$ s.t. $||K^j(t) - K^j(t - \Delta t)||_2 \leq C|\Delta t|,\ \forall\, \Delta t \in R,\ \forall j.$

• And lastly the shifted component functions satisfy a frame bound type of condition as follows:
$\sum_{k \neq i} \langle f_{p_i}(t - t_i), f_{p_k}(t - t_k) \rangle\ \leq \eta\ \forall\, i \in \{1, ..., N\}$

Then, reconstruction $X^*(t)$, resulting from the proposed framework, has a bounded noise to signal ratio. Specifically, the following inequality is satisfied:

$$||X(t) - X^*(t)||_2^2 / ||X(t)||_2^2 \leq (\delta + C\Delta)^{(1 + x_{max})} / (1 - \eta)$$

where $x_{max}$ is a positive number $\in [0, N - 1]$ that depends on the maximum overlap of the support of component functions $f_{p_i}(t - t_i)$.

**Proof:** A detailed proof of this theorem is provided in the appendix C.

## 6 EXPERIMENTS ON REAL SIGNALS

The proposed framework is general enough to apply to any class of signals. However, since the computational resources necessary to code and reconstruct video signals (function of three variables- $x, y, t$) would be sufficiently larger than audio signals (function of only one variable $t$), to demonstrate that the proposed framework can indeed be adopted in real engineering applications as a novel encoding scheme, we ran experiments on a repository of audio signals.

### 6.1 DATASET

We chose the Freesound Dataset Kaggle 2018 (or FSDKaggle2018 for short), an audio dataset of natural sounds posted on Kaggle referred in (7), containing 18,873 audio files annotated with labels from Google's AudioSet Ontology (9). For the purpose of the experiments, we ignored the labels and only focused on the sound data, since we were only interested in encoding and decoding the input signals. All audio samples in this dataset are provided as uncompressed PCM 16bit, 44.1kHz, mono audio files, with each file consisting of sound snippets of duration ranging between 300ms to 30s. In the experiment, we ran our proposed methodology over at least 1000 randomly chosen sound snippets from the samples in the dataset. For ease of computation, we kept the length of the input audio snippets to be relatively small (ideally of size less than 50ms), splitting longer signals. This choice of considering small snippets as input made the computation feasible on limited resource machines within reasonable time bounds by reducing the size of the $P$-matrix referred to in Equation 4. This choice is without loss of generalisation since for encoding signals of greater length, reconstruction using this framework can be done piece-wise: splitting a longer signal into smaller pieces, reconstructing piece-wise and finally stitching the reconstructed pieces together.

### 6.2 SET OF KERNELS

The proposed encoding technique is operational on a set of kernels, as stated in Equation 2. The first order of business was therefore the choice of a suitable set of kernels for our experiments. Since gammatone filters are widely used as a reasonable model of cochlear filters in auditory systems (14), and mathematically, are fairly simple to represent—$at^{n-1}e^{-2\pi bt}\cos(2\pi ft + \phi)$—in our experiments we chose a set of gammatone filters as our kernels (Figure 1). The implementation of the filterbank is similar to (16), and we used up to 2000 gammatone kernels whose center frequencies were uniformly spaced on the ERB scale between 20 Hz to 20 kHz. In all experiments, the kernels were normalized, and the baseline thresholds and the *ahp* parameters were kept the same across all kernels.

### 6.3 RESULTS

Following the assertion of Lemma 4, in all experiments, the baseline threshold and the absolute refractory period were kept low enough so that for each sound snippet near perfect reconstructions could be obtained at a high spike rate. A typical value of the refractory period was $\approx$ 5ms and the baseline threshold value was kept as low as $10^{-3}$. As a consequence of the corollary to Lemma 3, additional spikes did not hurt reconstruction. Experiments were conducted with varying number of kernels. Once a reconstruction at a high spike rate was attained, a greedy technique that removed spikes in order of their impact on the reconstruction was instituted to get a compressed code for each snippet. Reconstructions were then recomputed with the fewer spikes as constraints. We

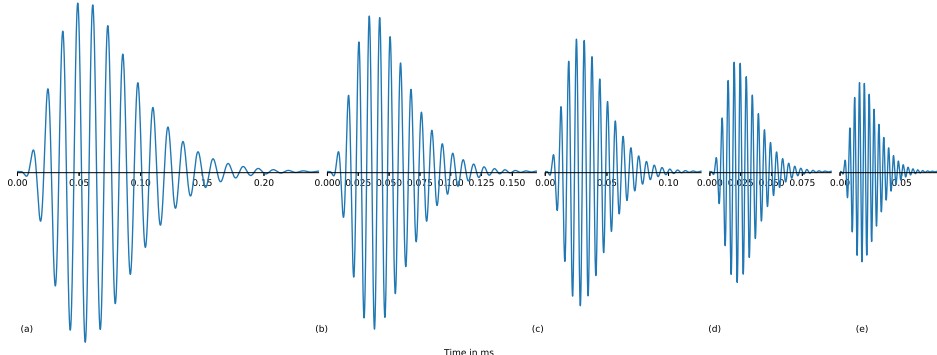

Figure 1: Five sample gammatone filters used as kernels with center frequencies located at approximately (a) 82 Hz, (b) 118 Hz, (c) 158 Hz, (d) 203 and (e) 253 Hz, respectively.

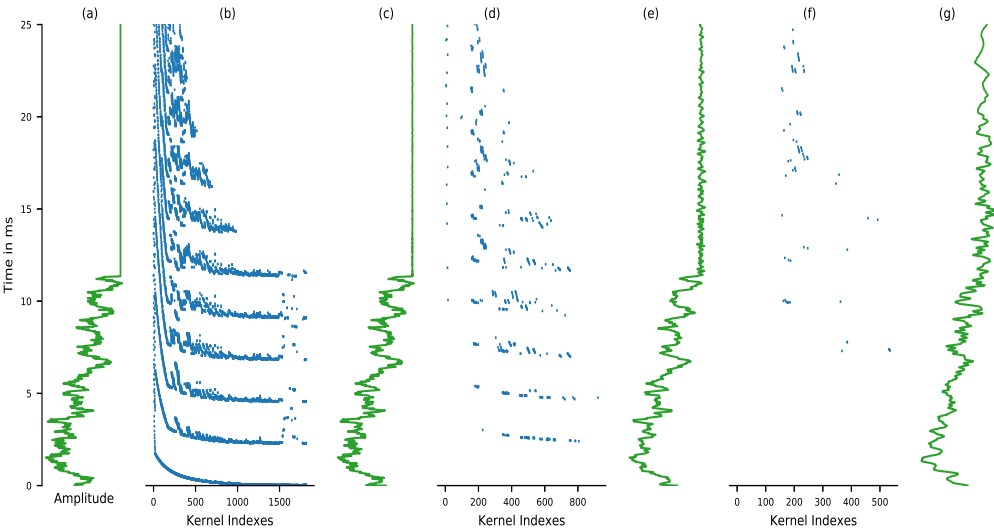

Figure 2: Reconstruction of a sample snippet in an experiment with 2000 kernels. (a) the input snippet, extended with zero padding to accommodate future spikes; (b) Spike trains of all kernels obtained at a low threshold and refractory period displayed as a raster plot with time (y-axis) and index of the gammatone kernels in increasing order of center frequencies (x-axis), and (c) the resulting reconstruction. This is an almost perfect reconstruction with a 32.7DB SNR at 1146 kHz spike rate of the ensemble. Subsequently spikes were deleted greedily (see text) to obtain reconstructions at lower spike rates. (d) Resulting spike pattern of the ensemble at 88.4kHz and (e) resulting reconstruction with SNR 19.7DB. Likewise, (f) spike pattern of the ensemble at 17.64kHz and (g) resulting reconstruction with SNR of 9DB. Time scale on left apply to all plots. It is noteworthy that in (d)&(f) spikes of the higher frequency kernels ended up deleted in the culling process.

should emphasize here that soon as spikes are removed to get a compressed representation of the signal, signals are no longer encoded via a simple *spike train* representation which ideally should communicate only spike times and not their corresponding threshold values. In other words, a compressed signal representation in this scheme needs to communicate both the spike times and the threshold values because once spikes are culled the decoder cannot infer the threshold values from the spike times and the given threshold function of the neurons in equation 1. In that sense a compressed representation of a signal in this approach can be realized through *marked spike trains* that carry both time as well as threshold information rather than a true *spike train* based representation. Figure 2 demonstrates this process applied to a sample sound snippet through several stages of removal of spikes. This process was repeated over 1000 randomly chosen sound snippets from the dataset. Figure 3 displays the complete results of the experiment with ≈ 2000 gammatone kernels. As the

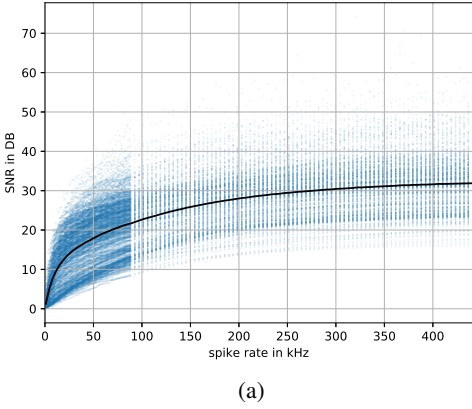
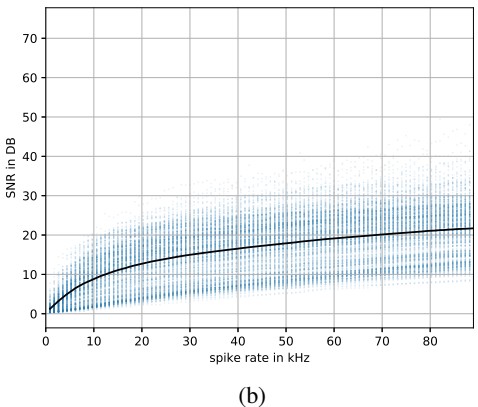

(a)                                                        (b)

Figure 3: Comprehensive result of an experiment with 2000 kernels reconstructing 1000 sound snippets. The graphs show scatter plots of reconstructions (each dot represents a reconstruction) with spike-rate of the ensemble (x-axis) against corresponding SNR value of the reconstructions (y-axis) (a) Scatter plot of reconstructions up to 441 kHz spike rate and (b) zoom in of the same graph for up to 89kHz. Solid line represents the average SNR of all reconstructions at a given spike rate.

figure demonstrates, at high spike rates nearly perfect reconstructions were obtained consistently, and even though lowering the spike rate gradually increased noise, reasonable reconstructions could be obtained at low spike rates ($\approx$ 15DB at 25kHz on average). Since each reconstruction is calculated by solving a system of linear equations involving a $P$-matrix whose dimension is $O(N^2)$ where $N$ is the number of spikes under consideration, computation is fairly time consuming, and the choice of parameters, such as the length of the input snippets, the number of kernels or the threshold parameters were made to ensure feasibility of computation with available resources while maintaining efficacy of the overall reconstruction process.

Since the proposed framework approximates a signal with a sparse linear combination of shifted kernels (as shown by signal class $\mathcal{G}$ in 4) and therefore has similarities to compressed sensing, another set of experiments were designed to compare the proposed framework with existing sparse coding techniques viz, Convolutional Matching Pursuit and Convolutional Orthogonal Matching Pursuit. Since the sparse coding techniques are computationally intensive over continuous signals, this set of experiments were restricted to only 10 gammatone kernels (for CMP and COMP all possible shifts of these 10 kernels were considered) and the experiments were run over 30 sound snippets. Our technique was applied as before, starting at a high spike rate with spikes culled gradually to achieve better compression. The results of comparison of average SNR values obtained by the techniques are shown in figure 4. As is evident, our technique in its simplest form does slightly better than COMP up until $\approx 50kHz$ beyond which COMP performs better. Our technique was $\approx 1.2$ times faster than COMP on an i7 hexa-core processor for these experiments and should naturally scale much better than COMP since the proposed technique does not involve repeated computation of inner products. The implementation details of the experiment can be found in our simulation code available at: `http://bitbucket.org/crystalonix/oldsensorycoding.git`.

## 7    Relation to Prior work

The problem of representing continuous time signals using ensembles of spike trains has a rich history both in the neuromorphic computing community as well as in computational neuroscience. Most such work rely on classical Nyquist–Shannon sampling theory wherein signals are assumed to be band-limited and reconstruction is realized through sinc filters, albeit via the spike trains. Among existing spike based coding techniques, (4) has explored the spike generating mechanism of the neuron as an oversampling, noise shaping analog-to-digital converter, and (11) represents signals via time encoding machines. Likewise, an image encoding technique has been discussed in (8) using an

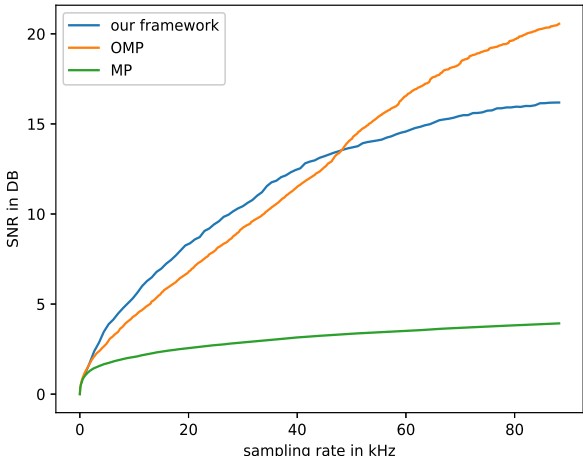

Figure 4: Average SNR values plotted as functions of sampling rate (spike rate for our framework and number of atoms chosen per second of the snippet duration in case of COMP/CMP).

integrate-threshold-reset framework that results in spike trains, which leverages differential pulse-code modulation (DPCM) at its core. In our case the input signals considered are elements of $L^2(\mathbb{R})$ with finite rate of innovation, the reconstruction error of which tends to zero as the signal approaches the span of appropriately chosen kernel functions which are again a generic class of continuous functions. Our work differs from existing approaches in that using our scheme signal reconstruction is realized via a sparse set of idealized spikes, whereas in the former case signals need to be sampled at a rate higher than the Nyquist rate and reconstruction implicitly relies on sinc interpolation. Since the class of signals $\mathcal{G}$ considered in our analysis takes the form $X(t) = \sum_{p=1}^{N} \alpha_p K^{j_p}(t_p - t)$, our problem formulation is comparable to that of convolution sparse coding or compressive sensing deconvolution, which, in general, is a hard problem and hence is solved under certain relaxed criteria (3) or by using certain greedy heuristics (12). Our framework provides a corresponding approximate solution to the general problem leveraging a biological thresholding scheme to produce spikes simultaneously at a high rate and then gradually removing unimportant spikes. The proposed technique, therefore, is a novel alternative to existing solutions.

## 8 CONCLUSION

We have proposed a framework that codes for continuous time signals using an ensemble of spike trains in a manner that is very different from the pulse-density paradigm. The framework applies to all finite rate of innovation signals, which is a very large class that includes bandlimited signals. Although approximate reconstruction is computationally more expensive than interpolation with a *sinc* kernel (as in Nyquist Shannon), it is feasible, unlike in the case of compressed sensing where the generic case is NP-hard. Fortuitously, the system of linear equations is best solved using the *conjugate gradient method* since $P$ is a symmetric positive semidefinite matrix. The excellent reconstruction results we have obtained with 2000 kernels—with no parameter tuning—is a testament to the potential of the technique. The human cochlear nerve, in comparison, contains axons of $\approx 50,000$ spiral ganglion cells (corresponding, therefore, to 50,000 kernels). As our theorems show, reconstruction with such a large set of kernels is guaranteed to be even better, albeit at a higher computational cost.

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

## A  PROOF LEMMA 2:

**Lemma 2:** Let equation 5 resulting from the optimization problem 2 have multiple solutions. Consider any two different solutions for $\alpha$, namely $\alpha_1$ and $\alpha_2$, and hence the corresponding reconstructions are given by $X_1(t) = \sum_{i=1}^{N} \alpha_{1i} K^{j_i}(t_i - t)$ and $X_2(t) = \sum_{i=1}^{N} \alpha_{2i} K^{j_i}(t_i - t)$, respectively. Then $X_1 = X_2$.

**Proof:** Let $S$ be the subspace of $L^2$-functions spanned by $\{K^{j_i}(t_i - t) | i \in 1, 2, ..., N\}$ with the standard inner product (by assumption each of $\{K^{j_i}(t_i - t) | i \in 1, 2, ..., N\}$ are $L^2$-functions and hence $S$ is a subspace of the larger space of all $L^2$-functions). Clearly $S$ is a Hilbert space with $dim(S) \leq N$. Hence there exists $\{e_1, ..., e_M\}$, an orthonormal basis of $S$ (where $M \leq N$). Assume that the hypothesis is false, i.e. $X_1 \neq X_2$. This implies that $\exists a_i$s and $b_i$s such that $X_1(t) = \sum_{i=1}^{M} a_i e_i$ and $X_2(t) = \sum_{i=1}^{M} b_i e_i$ where not all $a_i$s are same as the corresponding $b_i$s.
$\implies \exists k$ such that $a_k \neq b_k \implies \langle X_1(t), e_k \rangle = a_k \neq b_k = \langle X_2(t), e_k \rangle$
But $e_k \in span(\{K^{j_i}(t_i - t) | i \in 1, 2, ..., N\}) \implies \exists \{c_1, ..., c_N\}$ such that $e_k = \sum_{i=1}^{N} c_i K^{j_i}(t_i - t)$
$\implies \langle X_1(t), e_k \rangle = \sum_{i=1}^{N} c_i \langle X_1(t), K^{j_i}(t_i - t) \rangle = \sum_{i=1}^{N} c_i T^{j_i}(t_i)$
Now, since $X_1(t), X_2(t)$ are both solutions to the optimization problem 2
$\implies \langle X_1(t), e_k \rangle = \sum_{i=1}^{N} c_i \langle X_2(t), K^{j_i}(t_i - t) = \langle X_2(t), e_k \rangle$
$\implies \langle X_1(t), e_k \rangle = \langle X_2(t), e_k \rangle$        -which contradicts the hypothesis.        □

## B  PROOF OF LEMMA 4

**Lemma 4:** Let $X(t)$ be an input signal. Let $K^p$ be a kernel for which we want to generate a spike at time $t_p$. Let the inner product $\langle X(t), K^p(t_p - t) \rangle = I^p$. Then, if the baseline threshold of the kernel $K^p$ is $C^p \leq I^p$ and the absolute refractory period is $\delta$ as modeled in Equation 1, the kernel $K^p$ must produce a spike in the interval $[t_p - \delta, t_p]$ according to the threshold model defined in Equation 1.

**Proof:** The lemma is easily proved by contradiction. Assume that prior to and including time $t_p$, the last spike produced by kernel $K^p$ was at time $t_l$. Also assume that $t_l < t_p - \delta$ so that there is no spike in the interval $[t_p - \delta, t_p]$. Then by Equation 1 the threshold of kernel $K^p$ at time $t_p$ is $T^p(t_p) = C^p$. But, $\langle X(t), K^p(t_p - t) \rangle = I^p \geq C^p$. Furthermore, since $K^p(t)$ is a continuous function, the convolution $C(t) = \int X(\tau) K^p(t - \tau) d\tau$ varies continuously. By Equation 1, as the *ahp* kicks up the threshold to an arbitrarily high value $M^p$ at time $t_l$ and falls linearly to the value $C^p$ before time $t_p$, by the intermediate value theorem, the threshold must be crossed by the convolution $C(t)$ after time $t_l$ and before time $t_p$. Since $t_l$ was the last spike of kernel $K^p$ before time $t_p$, this is a contradiction. Hence, $t_l \not< t_p - \delta$, implying that there must a spike in the interval $[t_p - \delta, t_p]$.        □

## C  PROOF OF APPROXIMATE RECONSTRUCTION THEOREM

**Approximate Reconstruction Theorem:** Let the following assumptions be true:

- $X(t)$, the input signal to the proposed framework, can be written as a linear combination of some component functions as below:

$$X(t) = \sum_{i=1}^{N} \alpha_i f_{p_i}(t_i - t)$$

  where $\alpha_i$ are bounded real coefficients, the component functions $f_{p_i}(t)$ are chosen from a possibly infinite set $\mathcal{H} = \{f_i(t) | i \in Z^+, ||f_i(t)|| = 1\}$ of functions of unit $L^2$ norm with compact support, and the corresponding $t_i \in R^+$ are chosen to be bounded arbitrary time shifts of the component functions so that overall signal has a compact support in $[0, T]$ for some $T \in R^+$ and thus the input signal still belongs to the same class of signals $\mathcal{F}$, as defined in section 2.

- There is at least one kernel $K^{j_i}$ from the bag of encoding kernels K, such that the $L^2$-distance of $f_{p_i}(t)$ from $K^{j_i}(t)$ is bounded. Formally,
  $\exists\, \delta \in R^+$ s.t. $||f_{p_i}(t) - K^{j_i}(t)||_2 < \delta\, \forall\, i \in \{1, ..., N\}$.

- When $X(t)$ is encoded by the proposed framework, each one of these kernels $K^{j_i}$ produce a spike at time $t'_i$ at threshold $T_i$ such that $|t_i - t'_i| < \Delta\, \forall i$, for some $\Delta \in R^+$.

- Each kernel $K^j \in K$ satisfies a Lipschitz type of condition as follows:
  $\exists\, C \in R$ s.t. $||K^j(t) - K^j(t - \Delta t)||_2 \leq C|\Delta t|,\ \forall\, \Delta t \in R,\ \forall j$.

- And lastly the shifted component functions satisfy a frame bound type of condition as follows:
  $\sum_{k \neq i}\langle f_{p_i}(t - t_i), f_{p_k}(t - t_k)\rangle\ \leq \eta\, \forall\, i \in \{1, ..., N\}$

Then, reconstruction $X^*(t)$, resulting from the proposed framework, has a bounded noise to signal ratio. Specifically, the following inequality is satisfied:

$$||X(t) - X^*(t)||_2^2 / ||X(t)||_2^2 \leq (\delta + C\Delta)^{(1 + x_{max})}/(1 - \eta)$$

where $x_{max}$ is a positive number $\in [0, N - 1]$ that depends on the maximum overlap of the support of component functions $f_{p_i}(t - t_i)$.

**Proof of the Theorem:** By hypothesis each kernel $K^{j_i}$ produces a spike at time $t'_i\, \forall i \in \{1, ..., N\}$. Let us call these spikes as *fitting spikes*. But the coding model might generate some other spikes against $X(t)$ too. Other than the set of *fitting spikes* $\{(t'_i, K^{j_i})|i \in \{1, ..., N\}\}$, let $\{(\tilde{t}_k, K^{\tilde{j}_k})|k \in \{1, ..., M\}\}$ denote those extra set of spikes that the coding model produces for input $X(t)$ against the kernel bag K and call these extra spikes as *spurious spikes*. Here, $M$ is the number of spurious spikes. By Lemma1 $X^*(t)$ can be represented as below:

$$X^*(t) = \sum_{i=1}^{N} \alpha_i K^{j_i}(t'_i - t) + \sum_{k=1}^{M} \tilde{\alpha}_k K^{\tilde{j}_k}(\tilde{t}_k - t)$$

where $\alpha_i$ and $\tilde{\alpha}_k$ are real coefficients whose values can be formulated again from Lemma1. Let $T_i$ be the thresholds at which kernel $K^{j_i}$ produced the spike at time $t'_i$ as given in the hypothesis. Hence for generation of the *fitting spikes* the following condition must be satisfied:

$$\langle X(t), K^{j_i}(t'_i - t)\rangle = T_i \forall i \in \{1, 2, ..., N\} \tag{6}$$

Consider a hypothetical signal $X_{hyp}(t)$ defined by the equations below:

$$X_{hyp}(t) = \sum_{i=1}^{N} a_i K^{j_i}(t'_i - t), a_i \in R$$

$$\text{s.t.} \langle X_{hyp}(t), K^{j_i}(t'_i - t)\rangle = T_i, \forall i \tag{7}$$

Clearly this hypothetical signal $X_{hyp}(t)$ can be deemed as if it is the reconstructed signal where we are only considering the *fitting spikes* and ignoring all *spurious spikes*. Since, $X_{hyp}(t)$ lies in the span of the shifted kernels used in reconstruction of $X(t)$ using Lemma 3 we may now write:

$$||X(t) - X_{hyp}(t)|| \geq ||X(t) - X^*(t)|| \tag{8}$$

$$||X(t) - X_{hyp}(t)||_2^2 = \langle X(t) - X_{hyp}(t), X(t) - X_{hyp}(t)\rangle$$

$$= \langle X(t) - X_{hyp}(t), X(t)\rangle - \langle X(t) - X_{hyp}(t), X_{hyp}(t)\rangle$$

$$= \langle X(t) - X_{hyp}(t), X(t)\rangle - \Sigma_{i=1}^{N} a_i \langle X(t) - X_{hyp}(t), K^{j_i}(t - t'_i)\rangle$$

$$= ||X(t)||_2^2 - \langle X(t), X_{hyp}(t)\rangle$$

$$(\text{Since by construction} \langle X_{hyp}(t), K^{j_i}(t - t'_i)\rangle = T_i \forall i \in \{1...N\})$$

$$= \Sigma_{i=1}^{N} \Sigma_{k=1}^{N} \alpha_i \alpha_k \langle f_i(t - t_i), f_k(t - t_k)\rangle$$

$$- \Sigma_{i=1}^{N} \Sigma_{k=1}^{N} \alpha_i a_k \langle f_i(t - t_i), K^{j_k}(t - t'_k))\rangle$$

$$= \alpha^T F \alpha - \alpha^T F_K a \tag{9}$$

(denote $a = [a_1, a_2, ..., a_N]^T, \alpha = [\alpha_1, \alpha_2, ..., \alpha_N]^T$,

$F = [F_{ik}]_{N \times N}$, an $N \times N$ matrix, where $F_{ik} = \langle f_i(t - t_i), f_k(t - t_k) \rangle$

and $F_K = [(F_K)_{ik}]_{NXN}$ where $(F_K)_{ik} = \langle f_i(t - t_i), K^{j_k}(t - t'_k) \rangle)$

But using the results of Lemma1 $a$ can be written as:

$a = P^{-1}T$ where $P = [P_{ik}]_{NXN}, P_{ik} = \langle K^{j_i}(t - t'_i), K^{j_k}(t - t'_k) \rangle$

And, $T = [T_i]_{N \times 1}$ where $T_i = \langle X(t), K^{j_i}(t - t'_i) \rangle$

$= \Sigma_{k=1}^N \alpha_k \langle f_k(t - t_k), K^{j_i}(t - t'_i) \rangle = F_K^T \alpha$

$\implies a = P^{-1}F_K^T \alpha$

Plugging this expression of $a$ in equations 9 we get,

$$||X(t) - X_{hyp}(t)||_2^2 = \alpha^T F \alpha - \alpha^T F_K P^{-1} F_K^T \alpha \tag{10}$$

But,$(F_K)_{ik} = \langle f_i(t - t_i), K^{j_k}(t - t'_k) \rangle$

$= \langle K^{j_i}(t - t'_i), K^{j_k}(t - t'_k) \rangle$

$\qquad - \langle K^{j_i}(t - t'_i) - f_i(t - t_i), K^{j_k}(t - t'_k) \rangle$

$$= (P)_{ik} - (\mathcal{E}_K)_{ik} \tag{11}$$

(denoting $\mathcal{E}_K = [(\mathcal{E}_K)_{ik}]_{N \times N}$,

where $(\mathcal{E}_K)_{ik} = \langle K^{j_i}(t - t'_i) - f_i(t - t_i), K^{j_k}(t - t'_k) \rangle)$

Also, $(F)_{ik} = \langle f_i(t - t_i), f_k(t - t_k) \rangle$

$= \langle f_i(t - t_i) - K^{j_i}(t - t'_i) + K^{j_i}(t - t'_i),$

$\qquad f_k(t - t_k) - K^{j_k}(t - t'_k) + K^{j_k}(t - t'_k) \rangle$

$$= (\mathcal{E})_{ik} - (\mathcal{E}_K)_{ik} - (\mathcal{E}_K)_{ki} + (P)_{ik} \tag{12}$$

Combining 10, 11 and 12 we get,

$||X(t) - X_{hyp}(t)||_2^2 = \alpha^T F \alpha - \alpha^T F_K P^{-1} F_K^T \alpha$

$= \alpha^T \mathcal{E} \alpha - \alpha^T \mathcal{E}_K \alpha - \alpha^T \mathcal{E}_K^T \alpha + \alpha^T P \alpha$

$- \alpha^T P \alpha + \alpha^T \mathcal{E}_K \alpha + \alpha^T \mathcal{E}_K^T \alpha - \alpha^T \mathcal{E}_K P - 1 \mathcal{E}_K^T \alpha$

$= \alpha^T \mathcal{E} \alpha - \alpha^T \mathcal{E}_K P^{-1} \mathcal{E}_K^T \alpha$

$\leq \alpha^T \mathcal{E} \alpha$

$$\text{(Since, P is an SPD matrix, } \alpha^T \mathcal{E}_K P^{-1} \mathcal{E}_K^T \alpha > 0) \tag{13}$$

We seek for a bound for the above expression. For that we observe the following:

$|(\mathcal{E})_{ik}| = |\langle f_i(t - t_i) - K^{j_i}(t - t'_i), f_k(t - t_k) - K^{j_k}(t - t'_k) \rangle|$

$= ||f_i(t - t_i) - K^{j_i}(t - t'_i)||_2 ||f_k(t - t_k) - K^{j_k}(t - t'_k)||_2.x_{ik}$

(where $x_{ik} \in [0, 1]$. We also note that $x_{ik}$ is close to 0

when there is not much overlap in the support of the two

components and their corresponding fitting kernels.)

$\leq (||(f_i(t - t_i) - K^{j_i}(t - t_i)|| +$

$\qquad ||K^{j_i}(t - t_i) - K^{j_i}(t - t'_i))||).$

$\qquad (||f_k(t - t_k) - K^{j_k}(t - t_k)||$

$\qquad + ||K^{j_k}(t - t_k) - K^{j_k}(t - t'_k)||).x_{ik}$

$$\implies (\mathcal{E})_{ik} \leq x_{ik}.(\delta + C\Delta)^2 \tag{14}$$

Using Gershgorin circle theorem, the maximum

eigen value of $\mathcal{E}$:

$\Lambda_{max}(\mathcal{E}) \leq max_i((\mathcal{E})_{ii} + \Sigma_{k \neq i}|(\mathcal{E})_{ik}|)$

$$\leq (\delta + C\Delta)^2 (x_{max} + 1) \qquad \text{(Using 14)} \tag{15}$$

(where $x_{max} \in [0, N-1]$ is a positive number that depends on the maximum overlap of the supports of the component signals and their fitting kernels.)

Similarly, the minimum eigen value of $F$ is:

$$\Lambda_{min}(F) = min_i((F)_{ii} - \Sigma_{i \neq k} |\langle f_{p_i}(t - t_i), f_{p_k}(t - t_k) \rangle|)$$
$$\geq 1 - \eta \tag{16}$$

(By assumption $\Sigma_{i \neq k} | < f_{p_i}(t - t_i), f_{p_k}(t - t_k) > | \leq \eta$ )

Combining the results from 13, 15 and 16 we get:

$$||X(t) - X_{hyp}(t)||^2 / ||X(t)||^2 \leq \alpha^T \mathcal{E} \alpha / \alpha^T F \alpha$$
$$\leq \Lambda_{max}(\mathcal{E}) / \Lambda_{min}(F)$$
$$\leq (\delta + C\Delta)^2 (x_{max} + 1) / (1 - \eta) \tag{17}$$

Finally using 8 we conclude,

$$||X(t) - X^*(t)||^2 / ||X(t)||^2 \leq ||X(t) - X_{hyp}(t)||^2 / ||X(t)||^2$$
$$\leq (\delta + C\Delta)^2 (x_{max} + 1) / (1 - \eta)$$

