# OpenReview forum: "Signal Coding and Reconstruction using Spike Trains"
_ICLR.cc/2021/Conference — Reject_

### Official Review · AnonReviewer4 · 2020-10-20
**SIGNAL CODING AND RECONSTRUCTION USING SPIKE TRAINS**

**Rating:** 3
**Confidence:** 4

**Review:**

The authors describe a method for representing a continuous signal by a pulse code, in a manner inspired by auditory processing in the brain. The resulting framework is somewhat like matching pursuit except that filters are run a single time in a causal manner to find the spike times (which would be faster than MP), and then a N*N least squares problem is solved (which makes it slower). The authors claim that their method will perfectly reconstruct signals of finite innovation rate, however there appear to be mathematical errors in the proof.

First, while any function in the class G defined in section 4 has finite innovation rate, not all finite innovation rate functions are of this form. For example, suppose there was only one kernel K, which was band-limited, having no power above a frequency f. Then no signal with power above f can be represented as a finite sum of shifted copies of K (or even an infinite sum). Yet these signals can still have finite innovation rate (for example if band-limited to 2f).

Second, the use of the representer theorem appears to be incorrect. The L^2 norm defines a Hilbert space, but not a reproducing kernel Hilbert space as required for the representer theorem.  The representer theorem does not concern convolutions of L^2 functions with a kernel K, but an optimization over members of an RKHS whose norm is defined by K, whose objective function is defined by values at a finite set of points.

As a counter-example to the claimed perfect reconstruction theorem, consider the single boxcar kernel K(x) = {0 if x<0; 1 if 0<=x<=1; 0 if x>1}. Let the signal X(x) = K(x), which belongs to the class G with N=1, t_1 = 1, and alpha_1 = 1. Let the threshold T=0.5. Then the first spike will come at time 0.5; depending on the refractory period there may be any other number of spikes, but there will in general not be a spike at time 1. Thus the original signal is not in the class defined by equation (3).

In summary, this paper describes a potentially interesting biologically-inspired algorithm, but at least some of the claims appear to be incorrect.

---

> ### Author Response · Authors · 2020-11-13
> **Replies to Reviewer4's comments**
>
> We thank Reviewer4 for her/his careful read of our paper. Two of the points made arise from our failure to communicate clearly therefore leading to a misunderstanding, but the third (regarding RKHS) is due to our over-zealousness to cut down on space which we will rectify.
>
> **Use of Representer Theorem**
> This is indeed a very good catch which we will rectify. What we meant to say was "along the lines of the proof of the Representer theorem". The Reviewer is correct, the Representer theorem applies to RKHS where point evaluation can the written as a corresponding inner product, and therefore point evaluation constraints can be written as corresponding inner product constraints. The insight in the Representer theorem proof that the minimum squared norm solution should lie in the span of these vectors in the inner product constraints, is what we wanted to refer to. In our case, the shifted kernel convolutions are those inner products, and for the same reason as the Representer theorem, our theorem follows. To clarify even further, in our case also we can think of the $X^{*}(t)$ to be broken into two components- $X^{\parallel}(t)$, component in the span of $k^{j_i}(t_i-t)$s and $X^{\perp}(t)$, component in the space perpendicular to the space spanned by $k^{j_i}(t_i-t)$s. Since the component $X^{\perp}(t)$ does not affect constraints and to minimize the $L^2$ norm of the reconstructed signal, we let $X^{\perp}(t)$ go to zero and we get Equation 4. We are willing to modify this part to avoid any confusion.
>
> **FRI and band-limitedness**
> We fail to understand the Reviewers concern here. It is true that all band-limited functions are FRI functions since, as the Reviewer points out, they can be reconstructed using shifted sinc kernels placed at the Nyquist rate sample points (an $\Omega$-bandlimted function can be written as: $f(t) = \Sigma_{n \in Z} f(n\pi /\Omega)sinc(\Omega t-n\pi)$). The FRI class is indeed larger than bandlimited signals and our method applies to this class. What we have presented is the "most general" formulation of FRI, which, incidentally, is not our definition; it is pulled from the original TSP Vetterli et al paper (Eq 6 in their paper). Please note that the K's in the definition are general and they map to the phi's in Vetterli's definition. The overall claim of our paper is simply that if a signal is FRI then by definition there exists such K's, and if the neuron's convolution kernels are those K's then the signal can be reconstructed perfectly.
>
> **Counter-example to the Perfect Reconstruction Theorem**
> This is a great example which shows how model mismatch can hurt. We would like to clarify that $\textit{The Perfect Reconstruction Theorem}$ states that when the signal $X(t)$ belongs to the class $\mathcal{G}$ we can get perfect reconstruction only if we chose the time-varying threshold appropriately, i.e. if we can produce spikes at correct location (here at time t=1) as stated in the proof of the theorem and in Section 5 as well. Specifically, if we consider the reviewer's example of baseline threshold of 0.5 and if we use the threshold function defined in equation (1) and choose $M^j = 2$ and $\delta_j = 3/4$, or if we could simply set the baseline threshold at $T=1$ we would able to generate a spike at $t=1$ and we will get a perfect reconstruction in spite of the presence of spike at time $t=0.5$ (the coefficient corresponding to the spike at $t=0.5$ will go to zero). But we admit that in general we may not always get the spikes at the correct locations.
> In section 5, where we approach this problem from a more practical standpoint and try to analyze how exactly we could set the time-varying threshold so as to produce the spikes at the correct times $t_i$, we show in Lemma 4 that if we use the model definition of (1) we can actually generate spikes within time gaps of $\delta$ from the required times $t_i$. But then Reviewer#4 is right in his observation that this still does not generate spikes at exact times $t_i$ because in Lemma4 $\delta \neq 0$. That is why in section 5 we talk about approximate reconstruction. Specifically, we observe that the decoding is a continuous transformation between the spike trains to the reconstructed signal ($X^{*}(t) = \sum_{i=1}^{N} \alpha_i K^{j_i}(t_i-t)$) and hence a slight deviation in spike times would lead to only slight change in reconstruction error. But we agree that this claim could be backed by more rigorous math. For that we have already worked out the math which shows that even in case of a model mismatch one can show a bound on the error and have updated the paper. We would request our reviewers to kindly take a look at the Appendix A for the $\textit{Approximate Reconstruction Theorem}$. Since this theorem is too large to fit in the scope of this paper, we only talked about approximate reconstruction in an intuitive sense. We are willing to rephrase this part in a manner that conveys the key idea behind this in a much clearer manner.

---

### Official Review · AnonReviewer1 · 2020-10-23
**A new algorithm to encode continuous signals into spike trains, and to reconstruct the original signals from those spike trains.**

**Rating:** 7
**Confidence:** 3

**Review:**

PROS:
* new
* both theoretical and experimental results
* an alternative to matching pursuit, more accurate in certain regimes, which could have a broad range of applications

CONS:
* fully deterministic (see below)

The authors propose a new way to encode a broad class of continuous signals (those with finite rate of innovation, which include bandlimited signals) into spike trains, and to reconstruct those signals from the spikes. The reconstruction is exact under certain hypotheses: the signal should be a weighted sum of the kernels used by the neurons with some temporal shifts. In general several signals are compatible with a given spike train, and the algorithm finds the one with minimal energy.

This algorithm is an alternative to matching pursuit, and is shown to be more accurate in some cases (Fig 4).

In my opinion these results are new and worth sharing. I have only a few minor suggestions to improve the paper:
* How would the reconstruction degrade with noise in the spike trains, e.g. temporal jitter or extra/missing spikes?
* Noise in the (filtered) signal can be beneficial in some cases, because subthreshold signals can then cause spikes, with a higher rate if the denoised signal is near threshold. So the subthreshold signal can be estimated from the spike rates - something that would be impossible in the absence of noise. This is called stochastic resonance, and I wonder if the theory presented here could shed light on it.
* The author uses a soft refractory period, i.e. an adapting threshold, that is increased after a spike, and then goes back to the baseline linearly. From a biological point of view, an exponential decay would be more realistic. Could the theory cope with such an exponential decay?

MINOR POINTS:
* The abstract says "the transformation from external stimuli to
spike trains is essentially deterministic". This is highly debated!
* Eq 3 = Eq 4

---

### Official Review · AnonReviewer3 · 2020-10-26
**Signal Coding and Reconstruction using Spike Trains**

**Rating:** 5
**Confidence:** 4

**Review:**

In this paper, the authors describe a framework for stimulus encoding and reconstruction using spike trains. They characterise a class of signals where the proposed technique is guaranteed to yield perfect reconstruction. Finally, they provide examples of application to audio signals, and they compare their technique with existing sparse coding approaches.

Overall, the paper is convincing - the authors provide detailed theoretical grounding for their technique, and I wasn't able to spot any weakness from the formal standpoint. I only have one technical remark, and a few minor ones. Therefore, my initial recommendation is for the paper to be accepted.

### Technical remark:
From Lemma 1, stimulus reconstruction requires the values of the threshold functions at the time of spiking; spike times only are not sufficient. This information is typically not considered available in classic applications of spike coding, as a spike train is by definition just a set of spike times. This is not an obstacle to the practical application of the framework presented here, as one could always imagine transmitting the values of the threshold function along with the spike time when needed. On the other hand, it does make the claim that the framework operates with spike trains misleading, strictly speaking. I suggest that the authors highlight this in the introduction, title and/or abstract, for instance by talking about "marked" spike trains (by analogy to marked point processes), which carry additional information beyond the spike times.

###  Minor remarks:

1. Since this is a framework for stimulus reconstruction from spikes, it would have been good to acknowledge in the introduction the existing work on the same topic (though with different tools) done in the neuromorphic computing community. I encourage the authors to take a look at Gallego et al 2020 for a recent review.
2. At the beginning of Section 6: "the computational resources necessary to code and reconstruct (three dimensional) video signals is estimated to be an order of magnitude higher than (one dimensional) audio signals". Estimated by who? What exactly is meant here by "resources"? please provide references.
3. The annotations and text in the figures (labels, legends, etc) is so small as to be almost unreadable. Please make it larger.


###  References
G. Gallego et al., "Event-based Vision: A Survey," in IEEE Transactions on Pattern Analysis and Machine Intelligence, doi: 10.1109/TPAMI.2020.3008413.


-----
Post feedback edit:

The authors did not address the (admittedly, small) terminology issue I raised in my review. More specifically, I argued that their method in general requires transmission of the value of the threshold function, to which they replied that this is not true if the threshold function has a particular form (not required by their general theory), and then added some discussion about removal of spikes which was not what I was getting at. My point still stands: in general, for arbitrary threshold functions, transmission of the value of the function is needed for succesful decoding, and this constitutes more information than what is generally meant when discussing spike trains (which are just sets of time stamps).

Moreover, I was not convinced by the authors' reply to the points raised by Rev 4. This is actually more important than the point above, since Rev 4's objections are on the substance of the method (unlike my own points, which are little more than presentation and terminology matters).

Therefore, I am lowering my score.

I am not convinced by the authors' reply to Ref 4's comments. Moreover, the smaller terminology issue I raised in my own comment (about the

---

> ### Author Response · Authors · 2020-11-17
> **Replies to Reviewer3's comments**
>
> We are very grateful to Reviewer3 for the positive response to our submission. Reviewer3 has made certain relevant points that we respond to here.
>
> **Marked Spiked Trains**
> This is a great point made by reviewer3. In our initial encoding model, if we assume a certain form of the threshold function such as the one given in equation (1) and this threshold function is considered as $\textit{shared knowledge}$ between the encoder and the decoder, we can infer the threshold values from the spike times and the threshold function, and we need not communicate the threshold values explicitly to the decoder.  But it is very true that we need to convey both the threshold values and the spikes times to the decoder once we start removing spikes to get compact representations of signals as we have shown in our experiments. Since the core theory of this paper does not really appeal to such culling of spikes and is applicable only the experiment section, instead of pointing it out at the Introduction or elsewhere, we have incorporated this notable point at the experiment section right where we are really discussing removal of spikes. We would request our reviewers to consider this change made in experiment section where called out the compressed signals being represented via $\textit{marked spike trains}$.
>
> **Minor Points**
>
> *1. Existing Work*
>
> We thank the Reviewer for the pertinent reference. We have now added it to the introduction section.
>
> *2. Resources needed to code video signals*
>
> In this part what we meant was merely an order analysis for computation on our own resources. The experiments were run in parallel on 4 servers, each consisting of 2.3 GHZ, 10-Core Dual CPUs, 64GB of DDR4 RAM and 1TB of SSD storage. On these machines, the step-wise reconstruction process on each sound snippet starting from a high spike rate down to the lowest level, took≈20minutes on average. So for running similar experiments with one thousand video snippets on these machines would require much more time or we would need more powerful machines. We are willing to restate this part to avoid any confusion.
>
> *Size of annotations and texts in figures*
>
> We will take care of this.

---

### Official Review · AnonReviewer2 · 2020-10-29
**Interesting ideas but needs to clarify theory and contributions**

**Rating:** 3
**Confidence:** 4

**Review:**

This manuscript explores a mathematical framework and theory for a signal encoding/decoding scheme that shares many similarities to sparse deconvolution algorithms.  They demonstrate that they can learn a compressed representation of this signal, and there are several interesting theoretical properties.

Pros:
Some theoretical properties show intriguing properties
Seems to work well empirically

Cons:
The paper needs a significant revision for clarity.
I am not confident in all of the theoretical analysis.

I have some concerns about the theory.  First, there is a mismatch between the model definition in (1) and the theoretical analysis in Section 4.  The theory only holds if delta_j->0, which isn't discussed at all, and isn't what was used in the experiments. Or at least this is what is shown by the definitions given in (3) and (4).  This must be discussed at a minimum.

The "Perfect Reconstruction Theorem" is achieved by limiting the class to signals produced by the decomposition.  It is unsurprising that the system can decompose signals produced by it.  Equation 5 is fairly well-known from deconvolutional models.  The relationship to existing work needs to be more clearly defined, so that the contributions can be considered in context.

I'm confused by the claimed innovation in Lemma 3.  $X^*$ is defined at the minimum MSE for a number of spikes N.  Thus, it is the minimum energy and any other solution has equal or greater error.

Shouldn't equation 2 look at reconstruction error and not simply the minimum?

---

> ### Author Response · Authors · 2020-11-12
> **Replies to Reviewer2's comments**
>
> Thank you very much for your review!  We would like to clarify your concerns about our work.
>
> **Mismatch between the model definition in (1) and the theoretical Analysis and $\delta_j >0 $**
> Reviewer#2 has made an astute observation here. We want to clarify the point that the theoretical analysis in the Perfect Reconstruction Theorem holds for the most generic form of model with neurons having a "Time Varying Threshold"- $T^{j_i}(t)$, whereas definition (1) talks about a very particular kind of time varying threshold function in which due to the ahp effect the threshold shoots up to a very high value $M$ when the neuron spikes and then linearly falls back to the initial value within the refractory period and this simplified model is used utilized in section 5 and also in our experiments. On the other hand, the Perfect Reconstruction Theorem states that if a signal is of the form $X(t) = \sum_{i=1}^{N} \alpha_i K^{j_i}(t_i-t)$  where $K^{j_i}$ are in the bag of kernels, then if we can somehow get the corresponding kernels $K^{j_i}$ to spike at times $t_i$ we will get perfect reconstruction. This theorem does not make any assumption about the form of the time-varying threshold and thus does not particularly appeal to definition (1). In section 5, where we approach this problem from a more practical standpoint and try to analyze how exactly we could set the time-varying threshold so as to produce the spikes at those times $t_i$, we show in Lemma 4 that if we use the model definition of (1) we can actually generate spikes within time gaps of $\delta$ from the required times $t_i$. Reviewer#2 is meticulous in pointing out that this still does not generate spikes at exact times $t_i$ because in Lemma4 $\delta \neq 0$. That is why in section 5 we talk about approximate reconstruction. Specifically, we observe that the decoding is a continuous transformation between the spike trains to the reconstructed signal ($X^{*}(t) = \sum_{i=1}^{N} \alpha_i K^{j_i}(t_i-t)$) and hence a slight deviation in spike times would lead to only slight change in reconstruction error and that's why consistently our experiments show superior reconstruction even in practical scenarios. But we agree that this claim could be backed by more rigorous math. For that we have already worked out the math which shows that even in case of a model mismatch one can show a bound on the error and have updated the paper. We would request our reviewers to kindly take a look at the Appendix A for the $\textit{Approximate Reconstruction Theorem}$. Since this theorem is too large to fit in the scope of this paper, we only talked about approximate reconstruction in an intuitive sense. We are willing to rephrase this part in a manner that conveys the key idea behind this in a much clearer manner.
>
> **Innovation in Lemma3 and the objective of equation2**
> We are willing to rewrite parts of the Decoding section more clearly to avoid confusion about the objective of the equation2. For decoding the signal from its spike train representation we could not use the reconstruction error $||X(t)- X^*(t)||$, instead of the minimum energy signal as the objective function for a couple reasons: a) minimum reconstruction error $||X(t)- X^{*}(t)||$ objective will lead to the trivial solution which is $X(t)$ and b) also since we are trying to propose an encoding scheme, the objective of the Decoding module would be to reconstruct the original signal $X(t)$ just by looking at the spike data (times and threshold values) without accessing the input $X(t)$ , i.e. for the Decoder $X(t)$ is unknown. The innovation claimed by lemma 3 simply points to the fact that fortunately the chosen objective already gives a reconstruction which has the minimum reconstruction error in the span of shifted kernels generating spikes.
>
> **The Equation (5) and relationship with the existing work**
> As reviewer#2 pointed out, Deconvolutional  models feature something similar to our equation (5) which is because our underlying assumption is that the signal $X(t)$ is of form: $\sum_{i=1}^{N} \alpha_i K^{j_i}(t_i-t)$ and draws certain resemblances with sparse coding. But our approach is fundamentally different in the way the atoms (which in our case the shifted kernels producing spikes) are generated (via threshold crossing), unlike sparse coding our atoms are not chosen in a greedy manner and optimization is performed considering all spikes simultaneously. The underlying physical model as per our knowledge is novel. We are willing to incorporate these arguments in a separate section tagged as Related Work before the Conclusion.

---

### Comment · Area_Chair1 · 2020-11-18
**Discussion of prior work**

Dear authors,

could you please comment on how your work relates to prior work on reconstructing signals from spike trains, e.g.

Lazar A 2005, Multichannel time encoding with integrate-and-fire neurons
https://www.sciencedirect.com/science/article/pii/S0925231204004199?via%3Dihub

Lazar, A. A., and Pnevmatikakis, E. A. (2008). Faithful representation of stimuli with a population of integrate-and-fire neurons. Neural Comput. 20, 2715.
https://www.ncbi.nlm.nih.gov/pmc/articles/PMC3816090/

Your AC

---

> ### Author Response · Authors · 2020-11-18
> **Response to discussion of prior work**
>
> Dear Area Chair,
> We thank you for bringing up these relevant references. Please find below a detailed description of how these two papers relate to our work and how their work differs in its fundamental approach from ours.
>
> Admittedly, both the referenced papers [1,2] deal with an encoding model which amounts to a spike-train based representation of stimuli. Also, in [2] the encoding model performs convolution of the input signal against a bank of filters, followed by an integrate-and-fire model to produce spike trains, and hence the coding model is a superset of the standard leaky integrate-and-fire model. Our encoding model shows similarities to the referred model in this respect; our encoding model too considers a set of convolution filters prior to generation of spikes via thresholding (ours is a spike response model). But our approach differs from [1,2] fundamentally in the theoretical analysis of signal reconstruction and hence results in very different answers in terms of (1) the class of signals that the models can recover (2) the conditions for recovery and (3) the reconstruction algorithm itself. Specifically, in [2] the core theorem of the paper assumes that the signal is band-limited, the filters are considered to have spectral support which is a superset of the bandwidth and the lowest allowed spikes rate is greater than Nyquist rate. At this high spike rate perfect reconstruction follows from Jaffard's Lemma (referred in [2], Appendix C, Lemma1) which is a non-uniform version of the standard sampling theorem. Naturally, their reconstruction is sinc kernel based. In our theory, on the other hand, the class of signals considered for reconstruction is far more general (Finite Rate of Innovation) and does not appeal to band-limitedness and we don't impose any restrictions on the spike rate as such. Our reconstruction is not via the sinc kernel. Following is a high level description of the theoretical analysis presented in [2], which brings out how very different our attack on the problem is.
> Their framework begins with the assumption that the input $f(t)$ is a band-limited signal, i.e., $\hat{f}(\omega)$ has compact
> support $[-\Omega,+\Omega]$. One further assumes that there are $N$ linear filters (convolution kernels) $[h^1(t)$,$\ldots,h^N(t)]^T$ such that $h^j:{\mathbb R}\rightarrow {\mathbb R}$ has spectral support a superset of $[\Omega,+\Omega]$. The coding of the signal $f(t)$ proceeds as follows. First, $f(t)$ is convolved against each $h^j(t)$ to give $v^j(t)$, i.e., $v^j(t)= h^j(t)\circledast f(t)$. The signal $v^j(t)$ is then shifted up by a bias term $b^j$ to make it strictly positive, and then handed to an integrator, that is, $\int(v^j(t)+b^j)dt$. When the integrator reaches threshold $k^j$, a spike is generated and the integrator is reset to 0. The whole set up generates an ensemble of spike trains and corresponds to the constraints
> $
> %\begin{equation}
> \int_{t^j_i}^{t^j_{i+1}} ((h^j(t)\circledast f(t))+b^j) dt= k^j
> %\end{equation}
> $.
> The analysis of decoding from the spike trains can then leverage the sampling theorem. One uses the fact that both $f(t)$ as well as the $h^j(t)$'s can be represented as a weighted sum of sinc functions, to rephrase the above equations as inner product constraints in function space. Rates at which spikes have to be generated can then be determined based on requirements of perfect or approximate reconstruction, by appealing to the Nyquist sampling rate for $f(t)$.
>
> Intuitively speaking, since both $f(t)$ and $h^j(t)$ can be expanded as weighted combinations of shifted sinc functions, and since the convolution of two sinc functions is again a sinc function, each of the above integral constraint results in a linear constraint on the coefficients on the expansion of $f(t)$. So long as the spike rates are such that the system of such equations is either fully constrained or over constrained, the coefficients can be recovered exactly.
>
> We shall reference the two papers in our introduction in the part where we discuss related work.
>
> **References**
> [1]Aurel A. Lazar.   Multichannel time encoding with integrate-and-fire neurons.Neurocom-puting,  65-66:401 – 407,  2005.   ISSN 0925-2312. Computational Neuroscience: Trends in Research 2005.
>
> [2]Aurel A. Lazar and Eftychios A. Pnevmatikakis.   Faithful representation of stimuli with apopulation of integrate-and-fire neurons.Neural computation, 20(11):2715–2744, Nov 2008.ISSN 0899-7667.

---

### Decision · Program_Chairs · 2021-01-07
**Final Decision**

**Decision:**

Reject

**Comment:**

The paper presents a mathematical framework for encoding and decoding continuous valued signals from spiking neurons, proving both mathematical theory and simulation results. Two reviewers had serious concerns about the mathematical correctness and exactness of some of the presented mathematical results, and the AC raised questions about relationship to (un-cited) prior work. The authors aimed to address these shortcomings in the discussion phase, but neither me nor the reviewers were convinced that it provides a correct, and substantial, advance over prior approaches. I do hope that the feedback from the process will useful to the reviewers, and will help them in clarifying their contributions.

---

> ### Comment · ~Anik_Chattopadhyay1 · 2021-01-17
> **More specific comments would be appreciated**
>
> While we really appreciate the time and effort that has been spent in the review process and we thank all the reviewers, AC and PC for their valuable remarks, we believe that our research would be greatly benefited by more pin-pointed comments about what exactly the concerns are regarding the mathematical correctness of our proofs or about the comparisons made with related work. For example, reviewer4 had concern about the use of Representer theorem to which we acknowledged that it was a mistake in presentation and what we really meant was that the proof follows the same line of argument as in Representer theorem and rectified that problem. It must be noted that most of the proofs follow from simple linear algebra and the central theory of this paper has been tested through experiments run over years on real sound signals. Although we don't discard the possibility of mistakes, we believe our research would definitely benefit from more concrete statements as to what concerns the reviewers have regarding the correctness. Likewise, regarding the concern of the AC about similarity with Lazar's work we have clearly stated how our analysis differs fundamentally and explained that our analysis is far more general and has nothing to do with band-limitedness as in the work of Lazar et al, and we made necessary changes in the paper to cite Lazar's work.